

# Evaluation of the methane full-physics retrieval applied to TROPOMI ocean sun glint measurements

Alba Lorente[1], Tobias Borsdorff[1], Mari. C. Martinez-Velarte[1], Andre Butz[2,3], Otto Hasekamp[1], Lianghai Wu[1], and Jochen Landgraf[1]

[1]Earth science group, SRON Netherlands Institute for Space Research, Leiden, the Netherlands
[2]Institute of Environmental Physics, University of Heidelberg, Heidelberg, Germany
[3]Heidelberg Center for the Environment, University of Heidelberg, Heidelberg, Germany

**Correspondence:** Alba Lorente (a.lorente.delgado@sron.nl)

**Abstract.** The TROPOspheric Monitoring Instrument (TROPOMI) due to its wide swath performs observations over the ocean in every orbit, enhancing the monitoring capabilities of methane from space. In the short-wave infrarred (SWIR) spectral band ocean surfaces are dark except for the specific sun-glint geometry, for which the specular reflectance detected by the satellite provides a signal that is high enough to retrieve methane with high accuracy and precision. In this study, we build upon the
RemoTeC full-physics retrieval algorithm for land measurements and we retrieve four years of methane concentrations over the ocean from TROPOMI. We fully assess the quality of the dataset by performing a validation using ground-based measurements of the Total Carbon Column Observing Network (TCCON) from near-ocean sites. The validation results in an agreement of -0.5 ± 0.3% (-8.4 ± 6.3 ppb) for the mean bias and station-to-station variability, which show that glint measurements comply with the mission requirement of precision and accuracy below 1 %. Comparison to ocean measurements from the Greenhouse
gases Observing SATellite (GOSAT) satellite results in a bias of -0.2 ± 0.9% (-4.4 ± 15.7 ppb), equivalent to the comparison of measurements over land. The full-physics algorithm simultaneously retrieves the amount of atmospheric methane and the physical scattering properties of the atmosphere from measurements in the near-infrarred (NIR) and SWIR spectral bands. Based on the backscattering properties of the atmosphere and ocean surface reflection we further validate retrievals over the ocean. Using the "upper-edge" method, we identify a set of ocean glint observations where scattering by aerosols and clouds can
be ignored in the measurement simulation to investigate other possible error sources such as instrumental errors, radiometric inaccuracies or uncertainties related to spectroscopic absorption cross-sections. With this ensemble we evaluate the RemoTeC forward model via the validation of the total atmospheric oxygen ($O_2$) column retrieved from the $O_2$-A band, as well as the consistency of $XCH_4$ retrievals using sub-bands from the SWIR band, which show a consistency within 1%. We discard any instrumental and radiometric errors by a calibration of the $O_2$ absorption line strengths as suggested in the literature.

# 1 Introduction

Methane measurements over the ocean from the TROPOspheric Monitoring Instrument (TROPOMI) provide unprecedented monitoring capabilities for this potent greenhouse gas. Ocean measurements from satellite instruments in the shortwave-infrared spectral band are challenging due to low levels of reflected sunlight from water surfaces that may result in a low





signal-to-noise-ratio. For observations made under the specific sun glint geometry that provides specular reflection, the re-
flectance is strong compared to the off-glint dark ocean scenes. It provides a signal that is high enough to retrieve methane
concentrations that comply with the strict precision and accuracy mission requirements for atmospheric inversion analysis to
estimate methane sources and sinks.

TROPOMI is a nadir-viewing spectrometer that measures sunlight backscattered by Earth's surface and atmosphere. It uses
a push broom scanning technique and with its 2600 km wide swath, it obtains global coverage every day. The measurement
approach allows TROPOMI to perform observations under sun-glint geometry in every orbit. This is different from other
instruments like the one onboard the Greenhouse gases Observing SATellite (GOSAT) and the Orbiting Carbon Observatory-
2 (OCO-2) that use a pointing mechanism to observe scenes under sun-glint geometries, resulting in a reduced coverage
compared to that of TROPOMI. The sun-glint area (defined geometrically by the sun-glint angle in Eq. A1) depends on the
solar azimuth and zenith angles, resulting in a seasonal cycle on the global ocean coverage such that different latitudinal bands
are covered throughout the year (Fig. A1).

In order to retrieve methane from ocean sun-glint TROPOMI observations, we build upon the full-physics retrieval algorithm
applied to measurements over land that retrieves simultaneously the amount of methane in the atmosphere and the scattering
properties of aerosol and cirrus using both the near-infrared (NIR, 757-774 nm) and shortwave-infrared (SWIR, 2305-2385
nm) spectral bands. The full-physics retrieval uses the modification by aerosols of the strong absorption lines, the oxygen ($O_2$)
A-band in the NIR and methane ($CH_4$) and water ($H_2O$) in the SWIR, to obtain information about scattering in the atmosphere.
The RemoTeC algorithm has already been successfully applied to retrieve methane and carbon dioxide from sun-glint ocean
observations from GOSAT (e.g., Wu et al. (2020)) and OCO-2 (e.g., Wu et al. (2018)). Sources of systematic errors in the full-
physics retrievals of greenhouse gases include among others, instrumental errors and noise, spectroscopic uncertainties, and
inaccuracies in the characterization of scattering processes by aerosols and clouds in the forward model. Even though the full-
physics retrieval accounts explicitly for atmospheric scattering effects, the latter poses the biggest challenge for greenhouse
gas retrievals. It is therefore the error source that most extensively has been analysed in literature (e.g., Butz et al. (2009),
Houweling et al. (2005)). To account for it, usually empirical a posteriori bias corrections are applied to the retrieved methane
and carbon dioxide abundances.

For ocean observations, most of the light detected by the satellite in the sun-glint geometry comes from the direct reflection
of the solar backscattered light with very little contribution from diffuse backscattering, whereas over land single and multiple
scattering contribute substantially to the top-of-atmosphere reflectance. Therefore errors due to unaccounted light path modi-
fications by scattering processes for retrievals over ocean can be substantially different from those over land. The magnitude
and sign of these errors depend on the surface reflection, which is key to determine the effective optical path through the atmo-
sphere (Aben et al, 2007). Over land, both lightpath enhancement and shortening can happen. For scenes with moderate to high
surface reflectance, atmospheric scattering results in an enhancement of the lightpath due to multiple scattering, leading to an
overestimation of the trace gas if scattering is not properly accounted for. For scenes with low surface reflectance, atmospheric
scattering results mainly in a shortening of the lightpath as light might be scattered back before reaching the surface or absorbed





at the surface. Because a relatively strong scattering event is needed at the surface for lightpath enhancement, retrievals over the ocean mostly suffer from underestimation if scattering processes are not taken into account (Butz et al., 2013).

Based on the rationale explained above, Butz et al. (2011) proposed the so-called "upper-edge" method to identify an ensemble of ocean scenes with negligible scattering effects from which it is possible to discriminate between errors related to scattering processes and other sources like instrumental errors or spectroscopic uncertainties (Butz et al., 2013). The method assumes that when applying a retrieval based on an atmosphere with only molecular Rayleigh scattering, the retrieved oxygen (or surface pressure) is for most of the scenes underestimated, bounded by an upper-edge that corresponds to scenes where

aerosol scattering can be ignored. This method has been applied to several retrieval algorithms for GOSAT measurements (e.g., Butz et al. (2011); Crisp et al. (2012); Yoshida et al. (2013); Cogan et al. (2012)). These studies found a bias in the retrieved $O_2$ and surface pressure for the upper-edge ensemble and even though the cause of the bias remained unclear (e.g., Yoshida et al. (2013)), they all estimated a scaling factor of the $O_2$ column density of similar magnitude. This scaling factor was applied to the $O_2$ absorption line strengths, but because of the limited spatial coverage of the GOSAT data, the assessment of the scaling

factor effect on the retrieved greenhouse gas concentrations was limited in all these studies.

In this study we apply an optimized regularization scheme to the RemoTeC full-physics retrieval algorithm to obtain four years of methane concentrations over the ocean from TROPOMI sun-glint observations (Sect. 3) and we perform a full assessment of the quality of the dataset. First, the same way as for measurements over land, we validate sun-glint ocean measurements using ground-based measurements of the Total Carbon Column Observing Network (TCCON) network from near-ocean

sites (Sect. 4.1). We also compare methane from TROPOMI with retrievals from GOSAT (Sect. 4.2), and we evaluate the performance of the cloud and aerosol filter by analysing individual scenes for which dust outbreaks pose a challenge to the full-physics retrieval (Sect. 4.3). Due to the considerable amount of TROPOMI data over the ocean, we can further evaluate the retrievals over the ocean by applying the upper-edge method (Sect. 5). The purpose of applying the method is twofold: on the one hand, to diagnose the forward model for the $O_2$ A-band and validate the retrieved $O_2$, and on the other hand to obtain

an ensemble of scenes free of scattering effects to investigate other possible sources of error in the forward model.

## 2   Retrieval algorithm

We retrieve methane from TROPOMI with the RemoTeC full-physics algorithm, described in detail by Hu et al. (2016) and Lorente et al. (2021), as well as in the Algorithm Theoretical Basis Document (Hasekamp et al., 2021). The full-physics approach simultaneously retrieves the amount of atmospheric methane ($CH_4$) and the physical scattering properties of the

atmosphere from measurements ($y$) of sunlight backscattered by the Earth's surface and the atmosphere in the near-infrared (NIR, 757-774 nm) and shortwave-infrared (SWIR, 2305-2385 nm) spectral bands.

The forward model ($F$) employs the LINTRAN V2.0 radiative transfer model in its scalar approximation to simulate atmospheric light scattering and absorption in a plane parallel atmosphere (Schepers et al., 2014; Landgraf et al., 2001). In the forward model, surface reflection is modelled differently for land and ocean. For land scenes, surface is assumed to be Lamber-

tian, which means that light is reflected isotropically without accounting for the geometry-dependent properties of the surface





reflection. For ocean scenes, surface reflection is modelled using the wind speed dependent Cox-and-Munk reflection model (Cox and Munk, 1954) together with a wavelength dependent Lambertian term. Wind speed and surface pressure information are obtained from ECMWF operational analysis product and are therefore not retrieved.

The retrieval algorithm aims to find the state vector $\boldsymbol{x}$ that contains $CH_4$ partial sub-column number densities by solving the minimization problem:

$$\hat{\boldsymbol{x}} = \min_x \left( ||\boldsymbol{S}_y^{-1/2}(\boldsymbol{F}(x) - \boldsymbol{y})||^2 + \gamma ||\boldsymbol{W}(\boldsymbol{x} - \boldsymbol{x}_a)||^2 \right), \tag{1}$$

where $||\cdot||$ describes the Euclidian norm, $\boldsymbol{S}_y$ is the measurement error covariance matrix that contains the noise estimate, $\gamma$ is the regularization parameter, $\boldsymbol{W}$ is a diagonal unity weighting matrix that ensures that only the target absorber $CH_4$ and the scattering parameters contribute to its norm (Hu et al., 2016) and $\boldsymbol{x}_a$ is the a priori state vector. The retrieval state vector contains $CH_4$ partial sub-column number densities at 12 equidistant pressure layers. The total columns of the interfering absorbers CO and $H_2O$ are also retrieved, together with the effective aerosol total column, size and height parameter of the aerosol power law distribution as well as spectral shift and fluorescence in the NIR band. A Lambertian surface albedo in both the NIR and SWIR spectral range and its first-order spectral dependence is also retrieved. Because of the different treatment of surface reflection in the forward model for land and ocean scenes, the physical description of the surface albedo is different, and for ocean scenes it can lead to negative retrieved values. The final product of the retrieval is the methane total column-averaged dry-air mole fraction (XCH$_4$), calculated from the methane vertical sub-column elements $x_i$ and the dry-air column $V_{air,dry}$ calculated with meteorology input from ECMWF.

In Lorente et al. (2021), we optimized the regularization parameter $\gamma$ for the target absorber $CH_4$ and for each of the scattering parameters separately (effective aerosol distribution height and size parameter, and effective aerosol column). This resulted in a more stable performance of the inversion compared to the L-curve method in which the regularization strength changed at each iteration for every scene (Hu et al., 2016).

To infer XCH$_4$, ocean scenes are too dark except for the ones under the sun-glint geometry where the satellite observes the specular solar reflection at the ocean surface. In this geometry, direct solar backscattering from surface reflection dominates over the diffuse contribution of single and multiple scattering from aerosols. This makes the retrieval over ocean scenes less sensitive to aerosols than over land. Applying RemoTeC to ocean sun glint measurements with its configuration for land retrievals results in an unstable inversion for which particularly the retrieved aerosol height is dominated by noise. As the three effective aerosol parameters in the state vector are regularized separately with a constant regularization parameter for all the scenes (Lorente et al., 2021), we increase the regularization of the effective aerosol height distribution. Consequently, there is on average a 15% higher convergence rate over ocean than the previous choice of regularization parameters. Applying the same regularization over land has little effect (average global change of XCH$_4$ of -2.2 ± 6.44 ppb for one year of data). Therefore, we apply a consistent regularization both for land and ocean retrievals.



## 2.1 Posterior correction

Retrievals over land show an underestimation of $XCH_4$ for scenes with low surface albedo in the shortwave infrared spectral band. For these scenes, a posterior correction is applied based on the 'small area approximation' derived using only TROPOMI XCH$_4$. The validation with TCCON and GOSAT for land retrievals showed that the posterior correction reduces the regional bias by 6 pbb (Lorente et al., 2021).

For ocean and land scenes the forward model of RemoTeC is inherently different because the description of surface reflection is different (see Sect. 2), so land-ocean contrast in the retrieved $XCH_4$ is not unusual and there might be reasons to apply a different a posteriori bias correction for land and ocean observations. Retrievals over ocean do not show any correlation with signal or other retrieved parameters. So based on the global distribution of retrieved $XCH_4$ over ocean and over land we calculate a global correction factor. Even though most of the methane sources are located over land, a correction based on the annual median distribution of methane should not be affected by local emission events. To estimate the correction factor, we calculate the ratio between the median $XCH_4$ over land and over ocean. We compute the median separately for the Northern and Southern Hemisphere to account for the inter-hemispheric $XCH_4$ gradient. Here we limit the analysis to latitudes $60°N$ and $60°S$ as there are not sun-glint measurements at larger latitudes (see appendix A, Fig. A1 ). For the Northern Hemisphere the ratio of land and ocean median $XCH_4$ is 1.005 and for the Southern Hemisphere 1.003. We apply globally the average of the two, resulting in a factor of 1.004.

## 3 Retrieval results

We present four years of $XCH_4$ retrieved from TROPOMI measurements based on the retrieval approach explained in Sect. 2, including ocean measurements under sun-glint geometry. Figure 1 illustrates the global, yearly average TROPOMI $XCH_4$ both over land and over ocean. Year-to-year distribution reflects the increase in atmospheric $XCH_4$, depicted as well in the time series in Fig. 2 that shows annually averaged TROPOMI $XCH_4$ and surface dry air mole fraction of $CH_4$ from the marine surface network of the National Oceanic and Atmospheric Administration (NOAA) Earth System Research Laboratory. The TROPOMI global mean $XCH_4$ shows an annual increase with respect to previous year of 7.3, 10.6 and 15 ppb in 2019, 2020 and 2021 respectively. This tendency of increase in the growth rate that is also captured by measurements from marine surface sites, assuming full mixing in the atmosphere such that the trend in surface measurements and in the column measured by TROPOMI are comparable. Further breakdown of TROPOMI global average reflects the consistency between ocean sun-glint measurements and those over land, with a similar growth rate calculated with each of the measurements separately for 2018-2019 and 2020-2021 (see Table 1). For 2019-2020 a different growth rate is estimated using land (14.0 ppb) and ocean (8.5 ppb) measurements separately. This discrepancy may be due to a difference in coverage over ocean in 2020 and 2019 as a result of a bias in the cloud fraction used for filtering. The VIIRS Cloud Mask (VCM) used in the S5P-NPP processor of VIIRS data was biased towards higher cloud fractions in the center of the glint area over the ocean resulting in a reduced coverage in 2018 and 2019 (see Fig. 1). In March 2020 there was a switch in the cloud mask in favour of the Enterprise Cloud Mask (ECM) which no longer resulted in a bias in the cloud fraction. Once the VIIRS data is reprocessed with the ECM cloud mask for





155 2018 and 2019 (foreseen for fall 2022), the cloud fraction used to filter scenes and the coverage over the ocean will be uniform throughout the years.

**Figure 1.** Global TROPOMI XCH$_4$ distribution for (a) 2018, (b) 2019, (c) 2020 and (d) 2021, averaged in a cylindrical equal-area grid with a 0.3°x 0.5° grid at the Equator. TROPOMI XCH$_4$ with the bias correction applied is shown (see Sect. 2.1). Difference ocean coverage for 2018 and 2019 with respect to 2020 and 2021 is due to a bias in the VIIRS cloud fraction (see Sect. 3).

The spatial distribution along various coastlines like eastern South Africa and south-eastern Australia shown in Fig. 1 depict a slight gradient of XCH$_4$ between land and ocean close the coast. Further analysis of these gradients show that they are partly attributable to the computation of the yearly average that involves different seasonal coverage over land and ocean, 160 as high latitudes over the ocean are only covered partially throughout the year. On the other hand, as mentioned in Sect. 2.1, differences in the forward model between land and ocean may lead to residual differences in the retrieved XCH$_4$. This





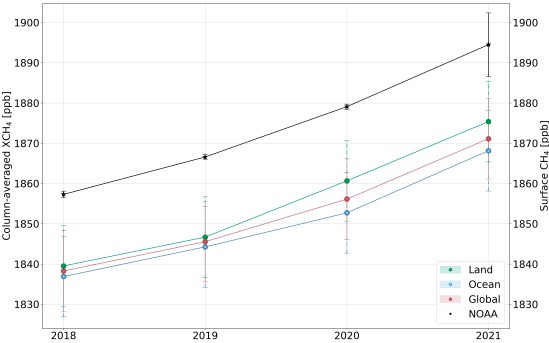

**Figure 2.** Time series of annual mean XCH$_4$ distribution calculated from TROPOMI measurements over land, ocean, and globally (land and ocean), and globally-averaged surface CH$_4$ determined from NOAA marine surface sites.

**Table 1.** Atmospheric XCH$_4$ growth rate calculated as the difference between annual mean column-averaged methane measured by TROPOMI over land, ocean and globally (land and ocean), and surface methane measured by NOAA marine ground-based sites.

|  | $\Delta$**XCH$_4$ [ppb]** | | |
|---|---|---|---|
|  | **2018-2019** | **2019-2020** | **2020-2021** |
| **TROPOMI Land** | 7.2 | 14.0 | 14.7 |
| **TROPOMI Ocean** | 7.4 | 8.5 | 15.4 |
| **TROPOMI Global** | 7.3 | 10.6 | 15.0 |
| **NOAA Surface** | 9.3 | 12.5 | 15.4 |

divergence is reduced by the posterior correction applied to XCH$_4$ retrieved over the ocean based on land retrievals (Sect. 2.1). Particularly over the eastern South Africa and south-eastern Australia coast the gradients are reduced from 1.0 - 1.5% to around 0.5%. The fact that land-ocean gradients are smaller than 1%, reflects the good agreement between land and ocean retrievals 165 and the fitness for purpose of the correction. Note that the estimation of the posterior correction factor is not affected by this spatial and temporal mismatch of the coverage since it involves averages based on multiple latitude ranges.

## 3.1 XCH$_4$ enhancements

TROPOMI XCH$_4$ observations over the ocean further contribute to the monitoring of CH$_4$ emissions thanks to the detection of strong XCH$_4$ enhancements. As an example, Fig. 3 shows single-pass TROPOMI XCH$_4$ measurements over eastern Turk-170 menistan and the Caspian sea, where multiple pixels with high XCH$_4$ enhancements are clearly visible. Close to the south coast TROPOMI detected two distinct plumes that originate in oil and gas facilities located inland from which satellite based emissions have already been reported (e.g., Varon et al. (2019)). On this specific day, the biggest enhancements are found in

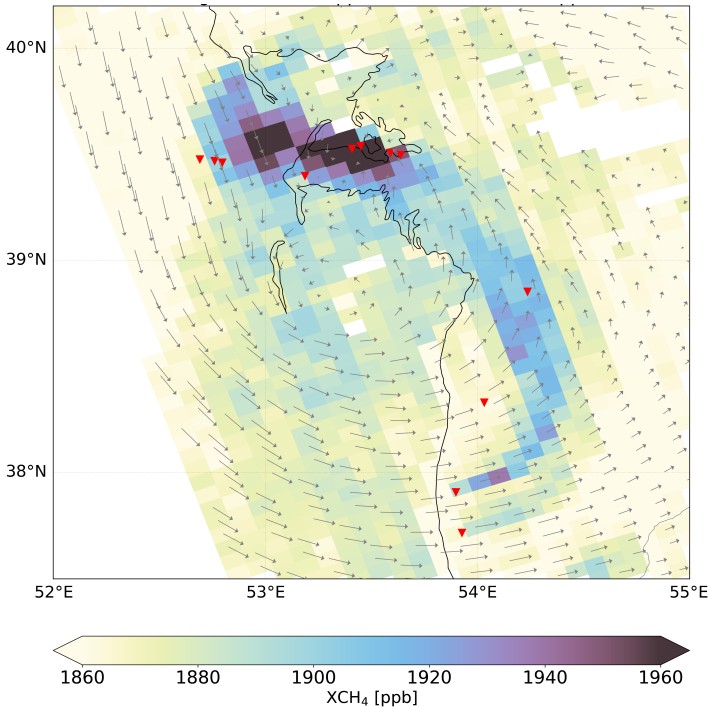

**Figure 3.** TROPOMI XCH$_4$ observations over Turkmenistan and the Caspian sea on 24 June 2018. Indicated with red triangles are inland oil and gas facilities and over sea locations where flaring was detected by VIIRS on this specific day. Wind arrows represent ECMWF 10m wind fields.

the Turkmenbashi Bay near multiple point sources (Irakulis-Loitxate et al., 2022) and over the ocean in front of the Cheleken peninsula close to locations where VIIRS detected flaring. While for the southernmost emissions relatively strong winds trans-

175 port the plume northward, in the northernmost area the low wind results in accumulation of air masses, therefore emitted CH$_4$ does not travel further resulting in XCH$_4$ enhancements as high as 120 ppb over a background of 1870 ppb.

The emission quantification methods typically applied to TROPOMI XCH$_4$ data (given its temporal and spatial resolution), like the cross-sectional flux method and the integrated mass enhancement, linearly depend on the detected XCH$_4$ enhancement with respect to the background value. For the particular example shown in Fig. 3, accounting for land pixels only results

180 in a total enhancement of 663 ppb, while ocean pixels make up to 1816 ppb of XCH$_4$ enhancement. Even if the detected enhancements over the ocean came from offshore emissions or transport from inland, the XCH$_4$ distribution and enhancement is homogeneous and not affected by land-ocean biases in the retrieval. This example highlights the importance of having observations over oceans to accurately quantify the total amount of CH$_4$ emissions.



# 4 Quality assessment

## 4.1 Validation with ground-based measurements

In this section, we validate the TROPOMI XCH$_4$ updated dataset (Sect. 3) with ground-based measurements from the Total Carbon Column Observing Network (TCCON) (Wunch et al., 2011a) (data version GGG2014, downloaded on 15 Dec 2021) for TROPOMI measurements over ocean for sun-glint geometries. We include the validation for measurements both over ocean and over land to put the results into perspective.

We use TROPOMI XCH$_4$ with a spatial collocation radius of 300 km around each station, and a temporal overlap of 2 hours for the ground-based measurement and the satellite overpass. We average TROPOMI XCH$_4$ and compare it to the TCCON XCH$_4$, and for all individual paired collocations we estimate the mean bias of TROPOMI-TCCON XCH$_4$ differences and its standard deviation. We then compute the average of the biases of all stations and its standard deviation as a measure of the station-to-station variability as a diagnostic parameter for the regional bias, following the approach in Lorente et al. (2021).

## Ocean

In order to validate TROPOMI XCH$_4$ retrievals over ocean, we select stations on islands or inland located close to the coastline (see Table B1) similarly to the validation of measurements from GOSAT and OCO-2 (e.g., Zhou et al. (2016)). Figure 4 shows the time series of XCH$_4$ measured by TROPOMI and TCCON for each of the stations for the period November 2017 - October 2021. TROPOMI XCH$_4$ sun-glint ocean measurements capture the seasonal variability and the year-to-year increase as also captured by TCCON XCH$_4$ measurements. This is true even for stations located close to the tropics (e.g. Burgos), where cloudiness challenges the coverage of TROPOMI XCH$_4$ measurements due to the strict cloud filtering. TROPOMI XCH$_4$ measurements show gaps due to the seasonality of the sun-glint geometry throughout the year. Overall, TROPOMI XCH$_4$ sun-glint measurements over ocean are biased low by -0.5% (-8.4 ppb), and the station-to-station variability is 0.3% (6.3 ppb). The effect of the posterior correction (Sect. 2.1) is to reduce the overall bias from -0.8% to -0.5%, without any effect on the station-to-station variability as it is a constant correction factor.

Izaña, located on the Canary Islands in Spain, is one of the stations that provides more recent data (see Fig. 4). This station is challenging due to dust outbreaks from the Sahara, and also because of the high altitude at which the TCCON instrument is located (2370 m.a.s.l.). For this high altitude station, TROPOMI measurements over ocean and those of the ground-based station look at significantly different atmospheric columns. Therefore, it is necessary to apply an altitude correction to avoid biases due to large height differences (e.g., Sha et al. (2021)). As Izaña measurements are always located at a higher altitude than the satellite measurements over ocean, the correction cuts off the partial column below the station height using the prior profile.





**Figure 4.** Time series of daily averaged XCH$_4$ measurements from TCCON (blue) and TROPOMI over ocean for sun-glint geometry (red) collocated with the selected stations for the period 1 Mar 2018 – 1 Oct 2021. TROPOMI measurements over ocean for sun-glint geometries around a circle of 300 km radius around each station have been selected for the validation.

**Land**

For the validation of measurements over land, we select the same 13 stations (Table B2) as in Lorente et al. (2021) for the results to be comparable between the different versions of the TROPOMI XCH$_4$ dataset. TROPOMI XCH$_4$ over land shows a





**Table 2.** Overview of the validation results of TROPOMI XCH$_4$ ocean sun-glint measurements with measurements from the TCCON network at selected stations. The table shows number of collocations, mean bias and standard deviation for each station and the mean bias for all stations and the station-to-station variability. Results are shown for TROPOMI XCH$_4$ with and without the posterior correction applied.

| Site, Country, Lat-Lon Coord. | Nr. of points | Corrected TROPOMI XCH$_4$ and TCCON | | Uncorrected TROPOMI XCH$_4$ and TCCON | |
|---|---|---|---|---|---|
| | | Bias [ppb] (%) | Standard deviation [ppb] (%) | Bias [ppb] (%) | Standard deviation [ppb] (%) |
| **Burgos** (Philippines) (18.52, 120.65) | 55 | −6.6 ( −0.4) | 16.7 (0.9) | −13.7 (−0.7) | 16.6 (0.9) |
| **Saga** (Japan) (33.24, 130.29) | 75 | 7.0 (0.4) | 11.6 (0.6) | −0.2 (0.0) | 11.6 (0.6) |
| **Tsukuba** (Japan) (54.36, −104.99) | 28 | −10.1 (−0.5) | 13.1 (0.7) | −17.2 ( −0.9) | 13.1 (0.7) |
| **Rikubetsu** (Japan) (43.46, 143.77) | 10 | −10.0 (−0.5) | 10.1 (0.5) | −17.0 ( −0.9) | 10.1 (0.5) |
| **Darwin** (Australia) (−12.46, 130.93) | 27 | −8.5 (−0.5) | 13.9 (0.8) | −15.5 ( −0.8) | 13.8 (0.8) |
| **Wollongong** (Australia) (−34.41, 150.88) | 14 | −8.2 (−0.5) | 11.1 (0.6) | −15.1 ( −0.8) | 11.1 (0.6) |
| **Reunion Island** (Australia) (−20.90, 55.48) | 19 | −4.0 (−0.2) | 10.7 (0.6) | −10.9 ( −0.6) | 10.7 (0.6) |
| **Izaña** (Canary Islands, Spain) (28.3, −16.5) | 97 | −13.0 (−0.7) | 13.0 (0.7) | −20.0 ( −1.1) | 12.9 (0.7) |
| **Edwards** (US) (34.95, −117.88) | 74 | −13.9 (−0.8) | 16.1 (0.9) | −21.0 (−1.1) | 16.0 (0.9) |
| **Pasadena** (US) (34.14, −118.13) | 65 | −17.2 (−0.9) | 13.7 (0.7) | −24.3 (−1.3) | 13.6 (0.7) |
| **Mean bias, station-to-station variability** | | −8.4 (−0.5) | 6.3 (0.3) | −15.5 (−0.8) | 6.3 (0.3) |

good agreement with TCCON measurements, with an overall bias and station-to-station variability of -0.3%. Even before the correction, the overall bias is within 1%. The posterior correction over land, which depends on the retrieved surface albedo, decreases the overall bias from -1.0% to -0.3% and the station-to-station variability from -0.6% to -0.3%, stressing the fitness for purpose of the correction. Based on the magnitude of the overall bias and station-to-station variability for land and ocean 220 retrievals, we can conclude that the data over land and ocean along the coast show a similar data quality.

## 4.2 Comparison with GOSAT satellite

In this section we compare TROPOMI XCH$_4$ with XCH$_4$ retrieved from measurements by the Thermal and Near Infrared Sensor for Carbon Observation Fourier transform spectrometer (TANSO-FTS) on board GOSAT. The GOSAT XCH$_4$ product is retrieved using the RemoTeC proxy retrieval, produced at SRON in the context of the ESA GreenHouse Gas Climate Change 225 Initiative (GHG CCI) project (Buchwitz et al., 2019, 2017).

We compare XCH$_4$ for the period 1 Mar 2018 - 31 Dec 2020, and we compute the average of daily biases and its standard deviation between TROPOMI and GOSAT measurements gridded in a 2° x 2° grid. Figure 5 shows XCH$_4$ retrieved from GOSAT (Fig. 5a) and TROPOMI (Fig. 5b). While over land the coverage of both satellites is relatively comparable after





**Table 3.** Overview of the validation results of TROPOMI XCH$_4$ land measurements with measurements from the TCCON network at selected stations. The table shows number of collocations, mean bias and standard deviation for each station and the mean bias for all stations and the station-to-station variability. Results are shown for TROPOMI XCH$_4$ with and without the albedo bias correction applied.

| Site, Country, Lat-Lon Coord. | Nr. of points | Corrected TROPOMI XCH$_4$ and TCCON | | Uncorrected TROPOMI XCH$_4$ and TCCON | |
|---|---|---|---|---|---|
| | | Bias [ppb] (%) | Standard deviation [ppb] (%) | Bias [ppb] (%) | Standard deviation [ppb] (%) |
| **Pasadena** (US) (34.14, −118.13) | 661 | −5.2 (−0.3) | 9.0 (0.5) | −1.0 (0.0) | 9.3 (0.5) |
| **Saga** (Japan) (33.24, 130.29) | 261 | 5.9 (0.3) | 14.8 (0.8) | −17.6 (−0.9) | 13.6 (0.7) |
| **Karlsruhe** (Germany) (49.1, 8.44) | 278 | −2.9 (−0.2) | 10.2 (0.5) | −19.6 (−1.1) | 10.5 (0.6) |
| **Darwin** (Australia) (−12.46, 130.93) | 187 | −11.0 (−0.6) | 13.3 (0.7) | −19.7 (−1.1) | 13.2 (0.7) |
| **Wollongong** (Australia) (−34.41, 150.88) | 412 | −8.4 (−0.5) | 11.7 (0.6) | −16.1 (−0.9) | 12.0 (0.7) |
| **Lauder II** (New Zealand) (−45.04, 169.68) | 357 | −2.6 (0.1) | 11.4 (0.6) | −16.3 (−0.9) | 11.1 (0.6) |
| **Park Falls** (US) (45.94, -90.27) | 555 | −9.0 (−0.5) | 14.3 (0.8) | −30.4 (−1.6) | 17.3 (0.9) |
| **East Trout Lake** (Canada) (54.36, −104.99) | 459 | −5.5 (−0.3) | 16.0 (0.9) | −28.6 (−1.6) | 18.1 (1.0) |
| **Lamont** (US) (36.6, −97.49) | 634 | −10.3 (−0.6) | 8.7 (0.5) | −17.5 (−0.9) | 9.6 (0.5) |
| **Orléans** (France) (47.97, 2.11) | 368 | −3.9 (0.2) | 11.7 (0.6) | −18.1 (−1.0) | 14.3 (0.8) |
| **Edwards** (US) (34.95, −117.88) | 748 | 0.9 (−0.0) | 8.9 (0.5) | 4.4 (0.2) | 9.4 (0.5) |
| **Sodankylä** (Finland) (67.37, 26.63) | 359 | −12.6 (−0.7) | 19.2 (1.0) | −42.1 (−2.3) | 19.4 (1.0) |
| **Mean bias, station-to-station variability** | | −5.4 (−0.3) | 5.1 (0.3) | −18.5 (-1.0) | 11.7 (0.6) |

performing the long-term temporal average and spatial gridding, over the ocean TROPOMI is able to cover a greater area than
GOSAT thanks to its wide swath. This makes the comparison of both XCH$_4$ datasets challenging, particularly over smaller
water bodies like Caspian Sea or the Mediterranean Sea.

Globally on average TROPOMI XCH$_4$ underestimates GOSAT XCH$_4$, as shown in Fig. 5c that depicts GOSAT to TROPOMI
XCH$_4$ ratio. Over ocean, the comparison leads to a bias and standard deviation of -4.4 ± 15.7 ppb (−0.2 ± 0.9 %). Before
the correction, the bias over ocean is -11.5 ± 15.7 ppb. Over land, the comparison leads to a bias after correction of -13.8 ±
16.1 ppb (−0.7 ± 0.8 %) and a Pearson's correlation coefficient of 0.87. The TROPOMI-GOSAT bias for land and ocean mea-
surements is more comparable before applying the correction over the ocean than after. This may be explained by the fact that
GOSAT may also have some land-ocean contrast in the XCH$_4$ distribution. As mentioned in Sec. 2.1, the inherent difference in
the forward model for land and ocean scenes results in residual differences between XCH$_4$ retrieved over land and over ocean.
While the correction applied to TROPOMI XCH$_4$ over the ocean aims to homogenize the XCH$_4$ distribution between land and





**Figure 5.** Global distribution of XCH$_4$ measured by (a) GOSAT and (b) TROPOMI and (c) the ratio of GOSAT to TROPOMI XCH$_4$. Daily collocations are averaged to a 2 x 2 degree grid for the period 1 March 2018–31 December 2020

ocean based solely on TROPOMI data, the correction applied to GOSAT ocean data is based on the validation with 4 TCCON stations (Buchwitz et al., 2019).

     The bias over land is of similar magnitude as before applying the updates on the retrieval algorithm to include ocean measurements (see Sec. 2) (Lorente et al., 2021). The standard deviation of the TROPOMI and GOSAT bias, as well as the station-to-station variability of the TCCON validation (Sect. 4.1) may be used as an estimation of the regional or spatially

variable bias to be applied in regional inversions that use TROPOMI XCH$_4$ data (e.g., Varon et al. (2022), Qu et al. (2021)).





### 4.3 Challenging scenes over ocean

Transport of dust towards sea surfaces from the Sahara and other desert areas are challenging for the full-physics retrieval of greenhouse gases. The desert dust particles are aerosols with high optical depth and with optical properties that may differ from the scattering properties assumed in the forward model. Inaccuracies in the modelling of the light path modifications produced

by scattering from aerosol particles over the relatively dark ocean surface may result in an underestimation of the retrieved $XCH_4$. This is visible in Fig. 1 in the Atlantic shore of the Saharan desert, as well as in the Red Sea and the Arabian Gulf. The current filtering process for the TROPOMI $XCH_4$ data related to scattering in the atmosphere consists in three steeps. First, VIIRS data is used to compute the cloud fraction based on the scenes classified as confidently and probably clear (Siddans, 2016), and we apply a threshold for the cloud fraction of 0.001. Then, the effective aerosol optical thickness (AOT) in the

near-infrared spectral band is used to filter scenes that can be affected by scattering, with a threshold of 0.3. The AOT in the short-wave infrarred can be applied as an equivalent filter with a threshold of 0.1, resulting in a more strict filtering. Finally, as a back-up option in case VIIRS data is not available, we apply a filter based on the differences in the retrieved $XCH_4$ and $H_2O$ between the strong and weak absorption bands (Hu et al., 2016).

Figure 6 shows an example of a single orbit where the sun glint area is affected by dust outflow from the Sahara. Figure

6a depicts retrieved $XCH_4$ after applying the VIIRS cloud filter. Figure 6b shows the MODIS corrected true color reflectance with the TROPOMI sun-glint area outlined in white, where a clear dust outflow event can be seen. The VIIRS data fails to filter out these scenes, as it classifies scenes as cloud-free, which not necessarily imply clear-sky. Figure 6c and d show the retrieved effective AOT in the NIR and SWIR spectral bands, with the color range adjusted to reflect the thresholds of 0.3 and 0.1 respectively. AOT distribution captures the high aerosol load both over land and over ocean. Over the sun-glint

area, the retrieved AOT captures the gradient of aerosol concentration, with a lower AOT just above 20°N latitude.The $XCH_4$ distribution over the sun-glint area is directly affected by the aerosols: between 15°N -20°N latitude band, the aerosol load acts as a highly reflective surface, resulting in a lightpath enhancement and an overestimation of the retrieved $XCH_4$. On the contrary, above 20°N latitude, the aerosol load is small compared to other areas, and over the relatively dark ocean scenes, the scattering effect results in a lightpath shortening and consequently in an underestimation of the retrieved $XCH_4$. Figure 6e and

f show the $XCH_4$ distribution after the filter based in the NIR AOT (0.3) and SWIR AOT (0.1) has been applied. The NIR AOT filter fails to filter the scenes with the strongest $XCH_4$ underestimation. The SWIR AOT filter is more stringent but few scenes with a clear $XCH_4$ underestimation still pass the filter.

This particular case highlights the difficulties in filtering the TROPOMI $XCH_4$ data when dust outbreaks cover scenes over the ocean, which are the most challenging scenes for the full-physics retrieval algorithm. The filter based on the SWIR AOT

is more strict than the NIR AOT; however, over land it is too strict and removes scenes where the retrieved $XCH_4$ is of good quality. Other TROPOMI products like the aerosol layer height (M. de Graaf et al., 2021) or the aerosol index (Stein Zweers, 2021) do not result in a more accurate filtering than the one based on the parameters from the $XCH_4$ retrieval. The use of other aerosol products from VIIRS for filtering dust events is currently under investigation.

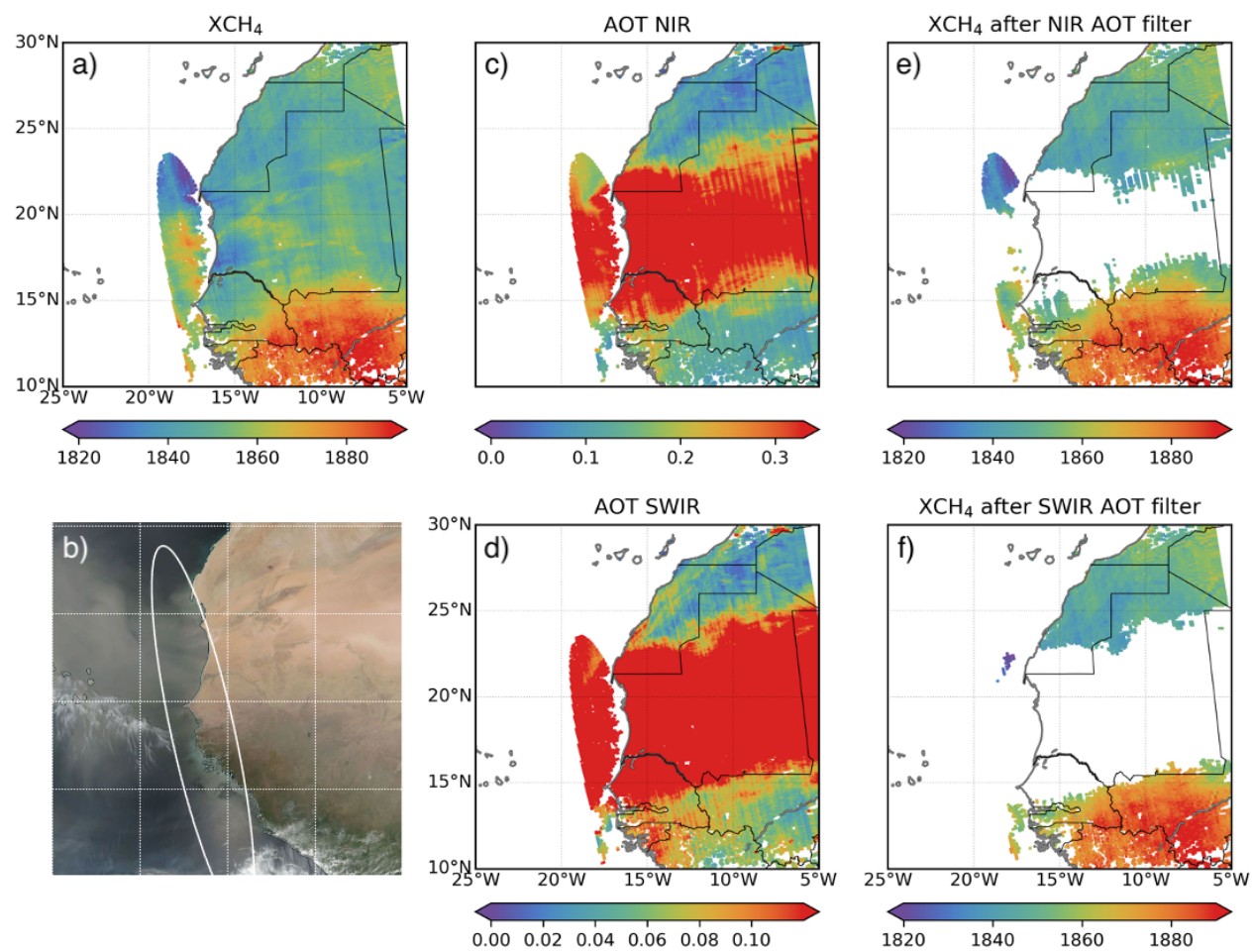

**Figure 6.** Single orbit (no. 7309) over ocean on 12 March 2019 (a) $XCH_4$ only filtered by clouds, (b) MODIS corrected true color reflectance, (c) NIR aerosol optical depth, (d) SWIR aerosol optical depth, (e) $XCH_4$ after aerosol filter (AOT NIR > 0.3), (f) $XCH_4$ after aerosol filter (AOT SWIR > 0.1). $XCH_4$ has been de-striped and corrected for surface reflectance spectral dependencies.





## 5   Upper Edge analysis

In the TROPOMI $XCH_4$ full-physics retrieval, the oxygen ($O_2$) A-band in the near-infrared spectral region provides part of the information on the full physics scattering properties of the atmosphere. The effect of atmospheric scattering on the $O_2$ A-band has already been discussed in detail in literature (e.g., Aben et al. (2007)). Ignoring atmospheric scattering by aerosols in this spectral band may lead to an under- or overestimation of the atmospheric light path in the forward model simulation, depending on the reflection at the surface and the scattering of light throughout the atmosphere. For glint observations, the

ocean surface is only bright for the particular geometry where the specular reflection occurs and dark for all other scattering geometries. This makes atmospheric multiple scattering of sun light between the surface and the atmosphere unlikely, favouring the shortening of light path due to single scattering of light by aerosols. Thus, total column $O_2$ retrievals from the $O_2$ A-band ignoring atmospheric scattering by aerosols results in an underestimation of retrieved $O_2$ due to unaccounted shortening of the light path through the atmosphere.

290       The particular scattering properties of the sun-glint observations are the basis for the so-called "upper-edge" method proposed by Butz et al. (2011, 2013) to validate GOSAT observations in the $O_2$ A-band. Total column $O_2$ retrievals from the $O_2$ A-band assuming a non-scattering atmosphere can be used to identify scenes over the ocean which are aerosol free. For this ensemble, the comparison of retrieved $O_2$ to total column $O_2$ concentrations derived from numerical weather prediction models (e.g., ECMWF) provide a validation tool, and can be used to investigate sources of error other than light scattering, like spectroscopy

uncertainties or instrumental errors. For this purpose, we retrieve $O_2$ total column for sun-glint ocean measurements by only accounting for molecular Rayleigh scattering and ignoring scattering by aerosols. We perform the analysis only for scenes that are cloud free based on measurements from VIIRS. We also apply a filter based on the non-scattering $H_2O$ and $CH_4$ retrieval from the weak and strong absorption bands (Hu et al. (2016), Lorente et al. (2021)) to remove scenes with clouds not captured by the VIIRS filter.

300       Figure 7a shows the time series of the weekly distribution of the ratio of retrieved $O_2$ to meteorological $O_2$ from ECMWF (European Centre for Medium-Range Weather Forecasts) operational atmospheric reanalysis (ERA5) product. For each week, we compute the histogram for ratio bins of 0.002, and normalize it to the maximum number of occurrences such that a value of 1 corresponds to the ratio bin with the highest value of occurrence (red in Fig. 7a). As we retrieve $O_2$ ignoring aerosol scattering effects in the atmosphere, the highest occurrence peak should be below the ratio of 1, which would imply an underestimation

of the retrieved $O_2$ due to the unaccounted scattering effects. However, the highest occurrence peak is found close to a ratio of 1.03. The upper-edge, which corresponds to scenes free of aerosol scattering effects, is defined as the $O_2$ ratio between $95^{th}$ and $99^{th}$ percentile of the distribution (following the selection criteria by Butz et al. (2013)). It is represented in Fig. 7a by the white dots. The statistical selection of the upper-edge is supported by the fit quality of the non-scattering $O_2$ retrieval (Fig. 7b), which coincides with a relatively low value of $\chi^2$ and low standard error for the upper-edge ensemble. An increase in $\chi^2$ and

its highest value corresponds to the scenes where scattering processes cannot be ignored, resulting in a poor quality of the fit and an underestimation of the retrieved $O_2$.



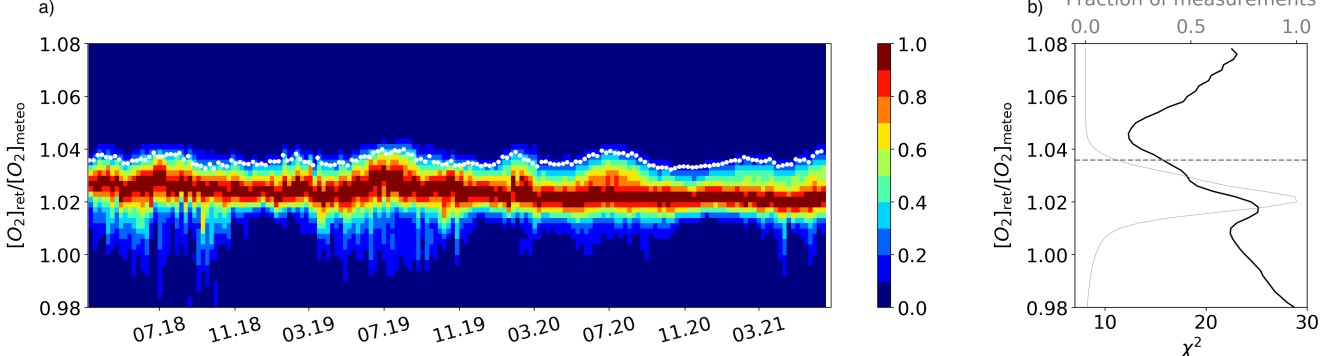

**Figure 7.** (a) Time series of weekly distribution of the ratio of retrieved non-scattering $O_2$ to meteorological (ECMWF) $O_2$. The ratio distribution is normalized such that the maximum occurrence corresponds to the red value 1. White dots are the upper edge ensemble defined as the scenes between the $95^{th}$ and $99^{th}$ percentile of the distribution. (b) Distribution of the $[O_2]/[O_{2,met}]$ ratio as a function of the fitting quality $\chi^2$ (black line) and the fraction of measurements for each ratio bins (grey line). Horizontal dashed line corresponds to the upper-edge factor computed as the average over the three-year period shown in (a).

The upper-edge should align with the $[O_2]/[O_{2,met}]$ ratio around the value of 1, that would mean that there are no scattering effects in the atmosphere and thus the non-scattering assumption in the retrieval is valid for these scenes. However, the average over the three-year period yields a factor of 1.0358 for the upper-edge. In March 2020 there was a switch in the VIIRS data used to calculate the cloud fraction for cloud filtering in the $XCH_4$ retrieval (see Sect. 3). Figure 7a shows that after that switch the distribution of the $[O_2]/[O_{2,met}]$ ratio is more uniform and less scattered. Data before March 2020 shows a more pronounced lower tail in the distribution, most likely because it contains scenes with some cloud contamination even after applying the filtering. The seasonal pattern throughout the complete time series of the distribution corresponds to the seasonality of the sun-glint area (see Appendix A). Between November and March the glint area covers mostly the Southern Hemisphere, and the distribution is narrower. Between March and November the glint area covers mostly the Northern Hemisphere, the lower tail is longer than in other periods, and the upper-edge shows a maximum in July. This is true even after the switch in the cloud mask, therefore the seasonality of the upper-edge is most probably caused by natural factors like dust outbreaks or residual cloudiness not detected by the filtering. Because of the relatively long time series available, neither the cloud mask switch nor the seasonality of the distribution affects the value of the upper-edge factor.

Previous studies that used the same method to evaluate $XCO_2$ and $XCH_4$ full-physics retrieval algorithms applied to GOSAT data (e.g. Butz et al. (2011); Crisp et al. (2012); Yoshida et al. (2013)), they all found a value of similar magnitude for the upper-edge factor (1.030, 1.025 and 1.01 respectively) which hints at potential non-instrumental errors with its source in the forward model. All these retrievals differ in the NIR spectral band used, in the forward model and in the definition of the state vector. As the upper-edge ensemble consists of scenes free of scattering errors, the bias in the retrieved surface pressure and $O_2$ in these studies was attributed to an underestimate of the spectroscopic $O_2$ A-band absorption cross sections such that it was introduced as a constant scaling factor to model the $O_2$ line strengths.



Following this approach, we introduce the upper-edge ensemble factor as a constant scaling factor in the $O_2$ cross section to model the $O_2$ absorption lines in the forward model and analyse how it affects the TROPOMI $XCH_4$ retrieval. Globally, based on one year of data from Sep 2018 to Sep 2019, the change in $XCH_4$ is negligible, with an average difference of 0.1

$\pm$ 3.7 ppb. The relatively high standard deviation of the differences (compared to the 0.1 ppb average), and a close to zero average difference resembles effects of opposite sign found at specific locations and periods. The effect of applying the $O_2$ scaling factor on retrieved $XCH_4$ is relatively more pronounced over high albedo scenes, like the Sahara Desert, and results in enhanced $XCH_4$ when including the $O_2$ cross-section factor. On average over North Africa (15N-30N, 20W-35E) retrieved $XCH_4$ is higher by 1.8 ppb (0.1%), with the highest differences found in January and February that locally can be up to 12 ppb

(0.6%) (the average for these months is 4 ppb (0.2%)). Over low albedo scenes, introducing the $O_2$ scaling factor results in lower retrieved $XCH_4$. On average over high latitudes over the Eurasian region (55N-75N, 50W-170W) the $O_2$ scaling factor lowers the retrieved $XCH_4$ by 3 ppb (0.15%) from February to April, with differences that locally can be up to 10 ppb (0.5%). For SWIR albedo between 0.15 and 0.2 the effect of the $O_2$ factor is smallest, which furthers supports the idea of an albedo range where errors in the quantification of light path modifications are minimum. The effect both for high and low albedo

scenes is a direct consequence of the enhancement and shortening of the light path by the $O_2$ scaling factor due to the physical effects of scattering in the atmosphere and absorption and reflection on the surface (e.g., Aben et al. (2007)). However, in any of the cases the fitting quality in the NIR band shows a clear improvement. The retrieved aerosol optical thickness in the NIR decreases on average from 0.1 to 0.09 when introducing the factor, with local differences that can be up to 0.03 but not necessarily for scenes where differences in $XCH_4$ are highest. This variable is used as a filtering parameter, but introducing the

$O_2$ scaling factor does not change significantly the number of scenes that pass the filter.

These findings show that the effect of the $O_2$ scaling factor on TROPOMI $XCH_4$ retrievals is overall not significant. In the specific cases where the change in retrieved $XCH_4$ is more pronounced, whether it results in an improved retrieval is not supported by the retrieval quality parameters, as it would be for example an improved fit quality. Butz et al. (2011) reported a higher $XCH_4$ by 0.26%, and a slightly bigger change in AOT than our results when the $O_2$ scaling factor was introduced

in the GOSAT retrievals. Yoshida et al. (2013) reported an improvement on the bias with TCCON for $XCH_4$ of 10 ppb. This differs from the present analysis, where TCCON validation does not capture any of the reported differences, neither does the comparison to $XCH_4$ retrieved from GOSAT measurements. Based on the effects reported in those studies compared to ours, it may be that the $O_2$ A-band has a higher effect on $XCH_4$ for GOSAT than for TROPOMI. In any case, these results suggest a different nature of the bias found when applying the upper-edge method, for which a simple scaling of the $O_2$ line strengths

might not be sufficient.

## 5.1 Sensitivity to fitting window

After we have defined the upper-edge ensemble, we investigate the consistency of the retrieved $XCH_4$ for different spectral windows in the shortwave-infrared spectral band for these scenes. Since the upper-edge ensemble should consist of scenes free of scattering effects, we expect good agreement in $XCH_4$ retrieved in the different fitting windows. To investigate this,

we process three months of data (March-May 2020, around 43500 scenes after filtering) for several fitting windows (see Table



4), using a non-scattering version of RemoTeC retrieval algorithm which ignores both Rayleigh and particle scattering in the atmosphere. As Rayleigh scattering is stronger for shorter wavelengths, the effect in the 2.3 $\mu$m band can be ignored.

**Table 4.** Spectral fitting windows used in the non-scattering XCH$_4$ retrieval for the upper-edge ensemble, and the bias and standard deviation of the differences of the retrieved XCH$_4$ with respect to the original fitting window W0.

| Window | Range [nm] | Interest | Bias w.r.t. W0 |
|---|---|---|---|
| W0 | 2307 - 2382 | Original fitting window | |
| W1 | 2307 – 2338 | Left side of spectrum | 10 ± 6 pbb<br>-0.6 ± 0.3 % |
| W2 | 2338 – 2382 | Right side of spectrum | 6 ± 5 pbb<br>0.3 ± 0.3 % |
| W3 | 2320-2338 | Strong absorption band (2317 nm) not included | 12 ± 14 pbb<br>0.6 ± 0.6 % |

The agreement in the retrieved XCH$_4$ for the various windows with respect to the original fitting window is on average within 0.6%. The bias and standard deviation of the differences is summarized in the last column of Table 4. The bias for the spectral window W1 which contains only the left side of the spectrum is negative, so retrieved XCH$_4$ is lower (-0.6%, -10 ppb), while for W2 which contains the right side of the spectrum it is positive and smaller in magnitude (0.3%, 6 ppb). These results are likely to be related to the different relative strength of the absorption bands that each of the fitting windows includes. For the spectral window W3, which ignores the strong absorption band at 2317 nm, the bias is positive (0.6%, 12 pbb) and similar in magnitude as for W1.

The differences in the retrieved XCH$_4$ for the different fitting windows are relatively small, but since these are presumably scenes free of aerosol scattering effects, we further investigate the possible sources of error that lead to such differences. For W1 and W2, the scatter in the differences is of the same magnitude as the bias or smaller; for W3 the scatter is largest in magnitude compared to W1 and W2 and slightly larger than the bias, and the effect of the measurement noise on the retrieval is also highest (not shown). This could be explained by the upper-edge ensemble containing scenes with residual scattering effects, which could affect more the narrower window that does not include the strong CH$_4$ absorption band. Another possible error source are the absorption line strengths and line shapes. The spectroscopy database for the absorption cross section used in the retrieval is the Scientific Exploitation of Operational Missions – Improved Atmospheric Spectroscopy Database (SEOM-IAS) (Birk et al., 2017). The agreement between the different fitting windows did not improve when substituting the absorption cross-section database by HITRAN 2008: the effect was to add an overall positive bias in the retrieved XCH$_4$, similar to what was found in Lorente et al. (2021). The evaluation of the quality of the spectral fit for the different windows did also not point to errors related to the calibration of the absorption lines. We also investigated the possibility of radiometric inaccuracies by fitting an intensity-dependent offset, which could not explain the difference in XCH$_4$ we observed.



## 6   Summary and conclusions

We have retrieved four years of $XCH_4$ from TROPOMI measurements over the ocean for sun-glint geometries, enhancing the

monitoring capabilities for methane with their addition to retrievals over land that are available since October 2017. We have optimized the S5P-RemoTeC full-physics retrieval algorithm for the specific information content on atmospheric scattering available from ocean measurements, where direct solar backscattering from surface reflection dominates over the diffuse contribution. Furthermore, the forward model has inherent differences on how land and ocean surface reflection are modelled. This results in residual differences between $XCH_4$ retrieved over land and ocean, so in order to homogenize the $XCH_4$ distribution

we have derived a constant correction factor for ocean retrievals independent of the retrieved parameters and based only on TROPOMI data. The TROPOMI global mean $XCH_4$ shows an annual increase with respect to the previous year of 7.3, 10.6 and 15 ppb in 2019, 2020 and 2021 respectively, a tendency of increase in the growth rate that is also captured by surface measurements.

The specular reflection of water surfaces in the specific sun-glint geometry results in a measured reflectance from which

methane can be retrieved with an accuracy and precision within the mission requirements. The validation with measurements from the ground-based TCCON sites from islands and near-coast sites results in a bias and station-to-station variability of -0.5 $\pm$ 0.3% (-8.4 $\pm$ 6.3 ppb), and the comparison to ocean measurements from the GOSAT satellite results in a bias of -0.2 $\pm$ 0.9% (-4.4 $\pm$ 15.7 ppb). Based on the magnitude of the bias, station-to-station variability and standard deviation for land and ocean retrievals, we can conclude that the data over land and ocean have a similar data quality.

Ocean measurements over sun-glint geometry provide an opportunity to further validate $XCH_4$ retrievals. The specific atmospheric backscatter properties and surface reflection over water surfaces allow to apply the upper-edge method on the $O_2$ A-band. The method rests on the principle that over ocean scattering processes always lead to an underestimation of the retrieved $O_2$ if lightpath shortening due to aerosol scattering processes is not taken into account. We use $O_2$ retrievals from the $O_2$ A-band to evaluate the forward model by comparing the retrieved $O_2$ to meteorological $O_2$. Based on the upper-edge method

we created a diagnostic dataset composed of scenes where scattering effects due to clouds and aerosols can be neglected such that it can be used to investigate sources of error other than approximations of scattering processes in the forward model. The $[O_2]/[O_{2,\text{met}}]$ ratio for these scenes free of aerosol scattering should align around the value of 1, however we find a bias of 1.04. The magnitude of this bias is similar to what previous studies found when applying the upper-edge method to GOSAT, which hints at potential non-instrumental errors. The hypothesis was that this bias may have its source in the forward model

such that the upper-edge factor was introduced as a constant scaling factor in the $O_2$ absorption lines strengths. By doing so in the TROPOMI $XCH_4$ retrievals, we conclude that the effect on $XCH_4$ is overall not significant, and that the sign and magnitude of the effect depends on surface albedo. Moreover, TCCON validation nor the comparison to $XCH_4$ retrieved from GOSAT capture any of the differences in retrieved XCH4 after applying the $O_2$ scaling factor. These results suggest a different nature of the bias found when applying the upper-edge method, for which a simple scaling of the $O_2$ line strengths might not be

sufficient. It also suggests that TROPOMI $XCH_4$ retrievals from the 2.3 $\mu$m are less affected by inaccuracies affecting the $O_2$ band than retrievals using the 1.6 $\mu$m band.





We have further used the upper-edge ensemble to investigate the consistency of the retrieved $XCH_4$ for different spectral windows in the shortwave-infrared spectral band using a non-scattering version of RemoTeC retrieval algorithm. The agreement in the retrieved $XCH_4$ for the various windows with respect to the original fitting window is of 0.8% on average, a consistency

within the precision and accuracy requirements of the mission. The different biases for the different windows are likely to be related to the different relative strength of the absorption bands that each of the fitting windows includes. The scatter of the differences is in all the cases similar or higher than the bias, which could be because the upper-edge may contain scenes with residual scattering effects. The evaluation of the quality of the spectral fit for the different windows did not point to errors related to the calibration of the absorption lines and we could not identify any radiometric inaccuracies.

*Data availability.* The TROPOMI $XCH_4$ dataset presented in this manuscript can be found at
https://ftp.sron.nl/open-access-data-2/TROPOMI/tropomi/ch4/18_17/. This corresponds to scientific SRON S5P RemoTeC TROPOMI $XCH_4$ product, which was processed as the pre-operational version of the operational TROPOMI $XCH_4$ retrieval. Measurements over ocean under sun-glint geometry with the retrieval settings presented in this manuscript were activated in the operational processing on November 2021, under version 2.3.0.

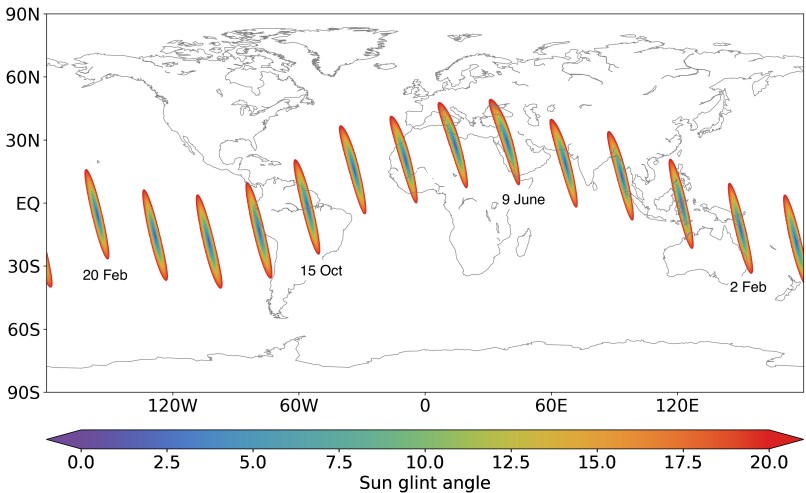

**Figure A1.** Sun glint angle seasonal cycle. Each coloured area corresponds to the sun glint area (i.e. SGA < 20°) for separate orbits in different months throughout the year, starting 1 Jan on the easternmost orbit and increasing months towards the West.

**Appendix A: Sun-glint measurement geometry**

The sun glint angle is defined as:

$$\alpha = \arccos(((cos(\theta_o + \theta) + cos(\theta_o - \theta)) + ((cos(\theta_o + \theta) - cos(\theta_o - \theta)) * cos(\phi_o - \phi))) * 0.5) \tag{A1}$$

Where $\theta_o$, $\theta$ are the solar and viewing zenith angle respectively, and $\phi_o$, $\phi$ are the solar and viewing azimuth angles. The dependency of the angle on the solar position results in a seasonal cycle on the coverage, i.e., different latitudinal ranges are covered throughout the year, as shown by the orbits plotted for different months in Fig. A1. Each glint area in the figure corresponds to one orbit for a specific month and day of the year. In a specific orbit, the sun glint area is defined such that the sun glint angle is lower than 20°. The signal measured over the ocean at scenes with low $\alpha$ (e.g. from 0° to 5°) can be twice as high as that measured at the edges (e.g. for $\alpha = 20°$. For values higher than 20°, the signal is too low and the noise dominates such that $XCH_4$ cannot retrieved with enough precision and accuracy.

**Appendix B: TCCON stations used for validation over land and ocean**

Here we summarize the geolocation of the TCCON ground-based stations used for validation of TROPOMI $XCH_4$ data in Sect. 4.



**Table B1.** Overview of the stations from the TCCON network used for validation of ocean sun-glint measurements in Sect. 4.1

| Site (Country) | Coordinates Lat, Lon (°) | Altitude (m.a.s.l.) | Reference |
|---|---|---|---|
| **Burgos** (Philippines) | 18.52, 120.65 | 40 | Velazco et al. (2017) |
| **Izaňa** (Canary Islands, Spain) | 28.3, -16.5 | 2370 | Blumenstock et al. (2014) |
| **Reunion Island** (France) | -20.90 55.48 | 90 | De Mazière et al. (2014) |
| **Rikubetsu** (Japan ) | 43.46 143.77 | 380 | Morino et al. (2016b) |
| **Tsukuba** (Japan ) | 54.36, -104.99 | 30 | Morino et al. (2016a) |
| **Pasadena** (US) | 34.14, -118.13 | 240 | Wennberg et al. (2017c) |
| **Edwards** (US) | 34.95, -117.88 | 30 | Iraci et al. (2016) |
| **Saga** (Japan) | 33.24, 130.29 | 10 | Kawakami et al. (2017) |
| **Darwin** (Australia) | -12.46, 130.93 | 30 | Griffith et al. (2017a) |
| **Wollongong** (Australia) | -34.41, 150.88 | 30 | Griffith et al. (2017b) |

**Table B2.** Overview of the stations from the TCCON network used for validation of land measurements in Sect. 4.1

| Site (Country) | Coordinates Lat, Lon (°) | Altitude (m.a.s.l.) | Reference |
|---|---|---|---|
| **Sodankylä** (Finland) | 67.37, 26.63 | 190 | Kivi and Heikkinen (2016) Kivi et al. (2017) |
| **East Trout Lake** (Canada) | 54.36, -104.99 | 500 | Wunch et al. (2017) |
| **Karlsruhe** (Germany) | 49.1, 8.44 | 110 | Hase et al. (2017) |
| **Orléans** (France) | 47.97, 2.11 | 130 | Warneke et al. (2017) |
| **Park Falls** (US) | 45.94, -90.27 | 440 | Wennberg et al. (2017a) |
| **Lamont** (US) | 36.6, -97.49 | 320 | Wennberg et al. (2017b) |
| **Pasadena** (US) | 34.14, -118.13 | 240 | Wennberg et al. (2017c) |
| **Edwards** (US) | 34.95, -117.88 | 30 | Iraci et al. (2016) |
| **Saga** (Japan) | 33.24, 130.29 | 10 | Kawakami et al. (2017) |
| **Darwin** (Australia) | -12.46, 130.93 | 30 | Griffith et al. (2017a) |
| **Wollongong** (Australia) | -34.41, 150.88 | 30 | Griffith et al. (2017b) |
| **Lauder*** (New Zealand) | -45.04, 169.68 | 370 | Sherlock et al. (2017) Pollard et al. (2019) |

*For the Lauder station the *ll* instrument was replaced on October 2018 to *lr*.

*Author contributions.* AL, TB, MCV, OH, AB and JL provided the TROPOMI CH$_4$ retrieval and data analysis. AL wrote the original draft with input from TB and JL. All authors discussed the results and reviewed and edited the paper.



*Competing interests.*   The authors declare that they have no conflict of interest.

*Disclaimer.*   The presented work has been performed in the frame of Sentinel-5 Precursor Validation Team (S5PVT) or Level 1/Level 2
Product Working Group activities. Results are based on preliminary (not fully calibrated or validated) Sentinel-5 Precursor data that will still
change. The results are based on S5P L1B version 1 data. Plots and data contain modified Copernicus Sentinel data, processed by SRON.

*Acknowledgements.*   The TROPOMI data processing was carried out on the Dutch National e-infrastructure with the support of the SURF
Cooperative. Funding through the TROPOMI national program from the NSO and Methane+ is acknowledged.



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
