# Peer review of "Evaluation of the methane full-physics retrieval applied to TROPOMI ocean sun glint measurements"

_Atmospheric Measurement Techniques, 2022_

## Author Comment (AC1)

**Response to the reviewers on the manuscript "Evaluation of the methane full-physics retrieval applied to TROPOMI ocean sun glint measurements" by Alba Lorente et al.**

The authors would like to thank the reviewers for their comments and suggestions. Below are the comments by the reviewers in blue and replies in black. Any modification made to the text has been underlined. The line and page numbers correspond to the version of the manuscript available for online discussion.
* * *
**Reviewer 1**

**Comment C 1.1** — Just one small detail that might need revision is the stated enhancements over the Turkmenistan OG facility site. In line 180, the total enhancements are stated to be 663 ppb (land) and 1816 ppb (ocean). I suspect these numbers refer to the sum of all per-pixel enhancements, but it might be nice to have that clarified with a few extra words (even though it is somewhat implicit by referring to the IME method).

**Reply**: We have modified the text accordingly: "accounting for land pixels only results in a total enhancement (i.e., sum of all per-pixel enhancements) of 663 ppb, while..."
* * *
**Reviewer 2**

**Comment C 2.1** — P2, L33. You can also mention that OCO-2 has a much narrower swath than TROPOMI and GOSAT has no swath, which is the main reason why their coverage is much poorer than that of TROPOMI.

**Reply**: We have included some more details on this topic: "The measurement approach allows TROPOMI to perform observations under sun-glint geometry in every orbit. This is different from other instruments that use a pointing mechanism to observe scenes under sun-glint geometries, resulting in a reduced coverage compared to that of TROPOMI. The instrument onboard the Greenhouse gases Observing SATellite (GOSAT) performs point measurements with a spatial resolution of about 10 km and the Orbiting Carbon Observatory-2 (OCO-2) obtains eight cross-track footprints of less than 1.29 km x 2.25 km resulting in a narrow swath of approximately 10 km."

**Comment C 2.2** — P2, L45. "the latter poses" -¿ "these pose"

**Reply**: Corrected

**Comment C 2.3** — P2, L46, "most extensively has been analysed" -¿ "has been analysed most extensively"

**Reply**: Corrected

**Comment C 2.4** — P2, L49-50, "direct reflection of the solar backscattered light". Is it really backscattered light? Isn't it direct light?

**Reply**: We have modified the text: "For ocean observations, most of the light detected by the satellite in the sun-glint geometry comes from the direct reflection of solar  light with very little contribution from diffuse scattering, ..."

**Comment C 2.5** — P2, L50 (and several other instances). You often use the word "backscattering" throughout the paper (e.g., "diffuse backscattering"), but is it really justified? If the instrument is looking at sun glint angles, I do not think it is ever looking in the backscattering direction (scattering angle of 180°). Wouldn't it be better to say "scattering"?

**Reply**: Thank you for bringing this up. Terminology and consistency is important. You are right that using backscatter is misleading because it refers to one specific type of scattering. We have removed the term backscattering throughout the text.

**Comment C 2.6** — P3, L64, "is for most of the scenes underestimated" -¿ "is underestimated for most of the scenes"

**Reply**: Corrected

**Comment C 2.7** — P4, eq. 1. I think this formulation is only valid if the measurement error covariance is diagonal. If you assume it is, explicitly say so when you explain the meaning of each term. If it is not, then $S_y^{-1/2}$ is not uniquely defined, and it would be more correct to write the first term on the right hand side as $(F(x) - y)S_y^{-1}(F(x) - y)$

**Reply**: Reviewer is right. We have added the term diagonal.

**Comment C 2.8** — P4, L97-99. Also point out that y is the vector containing the observations and F(x) is the output of the forward model in response to the state vector x.

**Reply**: We did not point this right after the equation because it was stated before in the first paragraph. We have in any case specified all the terms including those pointed by the reviewer after the equation to make it clearer.

**Comment C 2.9** — P5, L125. Can you briefly summarize what the "small area approximation" does, or at least add a reference? It may not be clear to the occasional reader.

**Reply**: We have modified the first paragraph of the section to explain a little bit more how the correction works and to add extra references. "[...] For these scenes, a posterior correction is applied based on the 'small area approximation' (O'Dell et al., 2018) which assumes a uniform $XCH_4$ distribution as a function of albedo in several regions around the globe. For each region, a $XCH_4$ reference value is estimated for a surface albedo around 0.2, and then the albedo dependency is obtained for all the regions combined (Lorente et al., 2021). The specific surface albedo value is selected because $XCH_4$ retrieval errors are lower in the SWIR for that albedo range (e.g., Guerlet et al., 2013; Aben et al., 2007).

**Comment  C 2.10**  —  P5, L145, "tendency of increase" -¿ "increasing trend"?

**Reply**: Corrected

**Comment  C 2.11**  —  P10, fig. 4. It may be good to report absolute bias, standard deviation and number of data points in each figure. Otherwise the figures look purely "illustrative".

**Reply**: We have included this information on the plots as suggested by the reviewer.

**Comment  C 2.12**  —  P14, L253-254. "First, VIIRS data is used ..., and we apply a threshold... then AOT is used...". I would advise to either use first or third person throughout.

**Reply**: Corrected

**Comment  C 2.13**  —  P20, L392, "solar backscattering from surface reflection". Is this not just "direct reflection of sunlight from the surface"?

**Reply**: Modified.

**Comment  C 2.14** — P20, L406, "atmospheric backscatter" -¿ "atmospheric scattering", "surface reflection over water surfaces" -¿ "reflection from water surfaces"

**Reply**: Modified.